# Polyphenol Composition and Antioxidant Activity of *Tapirira guianensis* Aubl. (Anarcadiaceae) Leaves

**DOI:** 10.3390/plants11030326

**Published:** 2022-01-26

**Authors:** Arnaud Patient, Elodie Jean-Marie, Jean-Charles Robinson, Karine Martial, Emmanuelle Meudec, Joëlle Levalois-Grützmacher, Brigitte Closs, Didier Bereau

**Affiliations:** 1Laboratoire COVAPAM, UMR Qualisud, Université de Guyane, 97300 French Guiana, France; nonomel@hotmail.com (A.P.); elodie.jean-marie@univ-guyane.fr (E.J.-M.); jean-charles.robinson@univ-guyane.fr (J.-C.R.); karine.martial@univ-guyane.fr (K.M.); 2UMR SPO, INRAE, Université de Montpellier, Institut Agro, 2 Place Viala, CEDEX 02, 34060 Montpellier, France; emmanuelle.meudec@inrae.fr; 3PFP, PROBE, INRAE, 34060 Montpellier, France; 4Campus Universitaire de Fouillole, Université des Antilles, 97110 Guadeloupe, France; levalois@inorg.chem.ethz.ch; 5Department of Chemistry and Applied Biosciences, ETH Zurich, 8093 Zurich, Switzerland; 6Entreprise SILAB, ZAC de la Nau, 19240 Saint-Viance, France; b.closs@silab.fr

**Keywords:** antiradical activity, polyphenol, UAE, DPPH, ORAC, TEAC, UPLC, fractionation, Amazonian plant

## Abstract

*Tapirira guianensis* (Anacardiaceae) is a natural resource from the Amazonian Forest and is locally known in French Guiana as “loussé” (creole), “tata pilili” (wayãpi), or “ara” (palikur). The tree is used by indigenous populations for medicinal purposes. To increase the potential of this tree for cosmetic, agro-food, or pharmaceutical uses, extracts were obtained through ultrasound-assisted extraction (UAE) from *T. guianensis* leaves using various extraction solvents such as water, methanol, and methanol–water (85/15; *v*/*v*). Chemical (DPPH, TEAC, ORAC) tests were applied to assess the anti-radical potential of these extracts. The polyphenol contents were determined by spectrophotometric (UV/Visible) and by means of chromatographic (UPLC-DAD-ESI-IT-MS^n^) methods. *Tapirira guianensis* leaf hydromethanolic extract produced the highest polyphenol content and exhibited antiradical activities in chemical assays (DPPH, TEAC, and ORAC) similar to (or higher than) those of a well-known antiradical plant, green tea. In *T. guianensis,* two classes of polyphenols were evidenced: (1) galloylquinic acids (identified for the first time in the studied species) and (2) flavonols and flavanols (present in small amounts). Flavonols seemed to play a major role in the antioxidant activity of DPPH. These findings provide a rationale for the use of *T. guianensis* in traditional medicine and to pave the way for seeking new biological properties involving this Amazonian tree.

## 1. Introduction

The use of plant-based extracts has become an important topic in recent years. Indeed, plants are known to contain important bioactive compounds that may be useful to guarantee the human well-being as well as for the preparation of supplements or nutraceuticals that are enriched with these compounds [1]. In this context, the use of plant leaves as alternative sources for the extraction of phytochemicals is a sustainable practice that has increased in recent years. Furthermore, in relation to the valorization of plants as sources of bioactive compounds, the dietary supplement market, and in general, that of nutraceuticals, is quickly growing.

Due to its wide distribution in central and South America, *T. guianensis,* locally known as “loussé” (creole), “tata pilili” (wayãpi), or “ara” (palikur), is used by indigenous populations for medicinal purposes: treatment for diarrhea, vomiting, bleeding, and vaginal infections; thrush young children; and as an anti-infectious agent (malaria, leprosis, syphilis, leishmaniosis) [2,3,4,5,6,7].

It is a medium-sized tree that grows as a source of secondary vegetation and becomes a large tree in primary forests. Thanks to its rapid growth (3 m height in 2 years and 14 m height in the final stage), the tree is used in Brazil in agroforestry and in the manufacture of toys, plywood, light wooden crates and boxes, inexpensive furniture, wood carvings, wooden soles, broom handles, etc. [8,9]. Although it is a common tree in French Guiana, this multifunctional source remains little used, and only the fruits are consumed. At present, there is no large-scale cultivation of *T. guianensis*, but new utilization applications (for making new colorless cosmetics and/or dietary supplements, for instance) may stimulate valuable activities, especially those involving leaves, which are so far considered agricultural waste.

*T. guianensis*, due do its polyphenol content, has been suspected to be involved in different biological properties. Anti-tumoral activities have been underlined in leaf extracts and we found to involve flavonoids (flavonols), tannins, and coumarins [10,11,12]. In rats, methanol extract (ethyl acetate fraction) from leaves of *T. guianensis* induced vasodilatory and myeloperoxidase-inhibitory were also pointed out in rats, it was determined that hydrolysable tannins (1,4,6-tri-*O*-galloyl-D-glucose, galloyl-HHDP-hexoside) and flavonoids (myricetin 3-O-α-L-rhamnopyranoside (myricitrin); quercetin 3-O-(6′′-O-galloyl)-β-D-galactopyranoside; quercetin 3-O-α-L-arabinofuranoside (avicularin); and quercetin 3-O-α-L-rhamnopyranoside (quercitrin) and quercetin) were involved [13,14]. Correia et al., 2008, also identified kaempferol 3-O-rhamnoside, kaempferol 3-O-arabinofuranoside, and kaempferol [15]. Phytochemical and biological studies have determined that the polyphenols (alkyl ferulates) present in the bark can lead to anti-bacterial, anti-protozoal, and anti-tumoral activities [15,16]. Gallic acid, quercetin, and quercitrin have also been demonstrated no cytotoxicity in flowers or no cytotoxicity when used in an adapted Brine shrimp lethality test [17].

Reactive oxygen species (ROS) and reactive nitrogen species (RNS) occur during oxidative stress and are responsible for many deleterious effects on living cells, as shown by many studies [18,19,20]. Such chemical entities are produced by the partial reduction of molecular oxygen and are most often free radicals. The effects of these toxic radicals can be reduced by free radical scavengers (FRS) such as polyphenols, which react with these entities to produce non-toxic (or at least, less toxic) molecules. Consequently, the consumption of FRS protects cells, slows the ageing process, and reduces the risk of pathological dysfunctions due to alterations in life-essential molecules. Polyphenols are among the most effective protective agents. To our knowledge, only one study has focused on the antioxidant activity of *T. guianensis.* Methanolic leaf extract and its fractions were investigated using DPPH as an AO test [13].

In our constant effort to develop natural resources from Amazonian plants and especially from those plants that are traditionally used by indigenous populations, we are always looking for new sources of bioactive compounds. Since no literature was found on AO assays implying that *Tapirira guianensis* leaf extracts were obtained using other solvents and methods and since no FRS contents (polyphenols, proanthocyanidins, flavonoids…) were described, this work intends to address this gap. To do this, the polyphenol and flavonoid content was determined spectrophotometrically (UV/Visible), and antioxidant assays (DPPH, TEAC, ORAC) were performed on *T. guianensis* leaf extracts obtained with water, methanol, and 85% methanol. The polyphenol composition of the latter extracts was also determined by chromatography coupled to diode array detection and electrospray ionization mass spectrometry (UPLC-DAD-ESI-IT-MS^n^).

## 2. Results and Discussion

### 2.1. Yield and Total Polyphenol Contents in Crude Tapirira guianensis Extracts

Three extraction solvents were used: pure MeOH, pure H_2_O, and 85% MeOH. The results are given in Table 1.

In Table 1, the extraction yields (Table 1) were similar (about 30%), regardless of the solvent that was used. As a greener alternative that involves the use of nontoxic chemicals and a better extraction yield of heat sensitive compounds, ultrasonic-assisted extraction and green solvents (water and methanol) were chosen [21]. The best results for the total polyphenol contents (TPC) were obtained using the methanol/water solvent (contents were twice as high as those obtained with methanol alone and three time higher than those obtained when used water alone). According to Chaves et al., 2020 [22], Polyphenols (almost flavonoids) are best able to be extracted from plant matrices through ultrasound-assisted extraction using polar and non-polar solvents (methanol, ethanol, acetonitrile, petroleum ether, acetone, water) and mixtures of these solvents.

Considering that there were no available data from previous studies concerning *T. guianensis (TG)*, our results were compared to black, green, and mate teas, as they represent polyphenol-rich plants. The total polyphenol contents from those tea samples were measured in pure methanol, 75% methanol, 50% methanol, and water extracts by Vural et al., 2020, using ultrasound-assisted extraction (UAE) [23]. TPC in the aqueous extracts of *TG* a were better than those obtained from black tea (29.6 ± 0.97 mg GAE/g DM) but less than those obtained the green and mate teas (50.72 ± 1.99 mg GAE/g DM and 56.38 ± 2.72 mg GAE/g DM, respectively). Concerning the TPC in pure methanolic extract, only green tea (78.98 ± 0.43 mg GAE/g DM) had a better value. Dealing with the hydromethanolic extract, TPC was better than all of the studied tea samples. Thus, this extract was used for chromatographic analyses.

The total flavonoid contents (TFC) were higher in the methanol extract than they were in water and 85% MeOH (1.5 times). Compared to the black, green, and mate teas, the TFC in the *TG* aqueous extract was higher than it was in black tea (11.45 ± 0.20 mg GAE/g DM) but lower than it was in the green and mate teas (23.76 ± 1.26 mg GAE/g DM and 30.94 ± 1.00 mg GAE/g DM, respectively) [23]. The same trend was observed concerning the *TG* pure methanolic extracts (14.24 ± 1.02 mg GAE/g DM; 28.45 ± 1.31 mg GAE/g DM and 56.80 ± 2.11 mg GAE/g DM for black, green, and mate teas, respectively). Flavanols were more present in the methanol and methanol/water extracts (three times more than in water), but their amounts were very low.

### 2.2. UPLC-DAD-ESI-IT-MS^n^ Analysis of the 85% MeOH Extract

Figure 1 shows the 16 major peaks that were observed in the chromatograms recorded at 280 and 360 nm.

Mass fragmentation data are given in Table 2.

These compounds were tentatively identified on the basis of their retention times, UV spectra characteristics, and MS data in the positive and negative ion modes presented in Table 2.

Compounds **1** to **4** displayed the UV spectra characteristic of phenolic acids. The mass spectra in the negative ion mode (ESI^-^) showed pseudo-molecular ions [M−H]^−^ at m/z 495, 495 and at 647 and 799, respectively. Compounds **1** and **2** are isomers yielding parent ions at m/z 495 and m/z 497 in negative and positive ion mode, respectively. According to Clifford and collaborators, their molecular weight (MW) corresponds to digalloyl quinic acids [24]. In the negative mode, the MS^2^ base peaks at m/z 343, indicating the loss of one galloyl residue (152 amu). Other secondary fragments were produced from compound **1** at m/z 325 (dehydrated fragment; 2% of base peak), m/z 191 (3%), and m/z 169 (6%). All of these fragmentations were consistent with 3,5-digalloylquinic acid. The secondary MS^2^ fragments from compound **2** were more intense at m/z 343 (base peak), m/z 325 (38% of base peak), m/z 193 (18%), and m/z 169 (14%). This could fit with 4,5-digalloylquinic acid. In positive ion mode (ESI^+^), corresponding molecular ions [M+H]^+^ of compounds 1 and 2 were detected at m/z 497. The MS^2^ base peaks were due to a loss of water (m/z 479) in both compounds. The MS^3^ fragmentation of the peak at m/z 479 for both compounds produced an ion at m/z 309 due to a gallic acid loss.

Compound **3** produced a molecular ion at m/z 647 in the negative ion mode and a major MS^2^ fragment m/z 495 (base peak), which could be attributed to the loss of a galloyl residue (152 amu) due to the secondary fragments at m/z 477 (13%; −18: loss of H_2_O), m/z 343 (12%; 477 − 152: loss of a second galloyl residue), and m/z 325 (7%, −18: loss of H_2_O). Such fragmentation is consistent with data published by Clifford et al., 2007, for 3,4,5-trigalloylquinic acid [24]. In the positive ion mode, a molecular ion [M+H]^+^ appeared at m/z 649. The loss of one gallic acid was the only fragmentation that occurred in the MS^2^ spectrum (m/z 479). A new gallic acid loss produced a MS^3^ base peak at m/z 309. Therefore, these results confirmed the negative mode hypothesis.

Compound **4** was tentatively identified as a 1,3,4,5-tetragalloylquinic acid according to Clifford et al., 2007 [24]. In the negative ion mode, a molecular ion [M−H]^−^ was produced at m/z 799. MS^2^ fragmentations yielded a base peak at m/z 601 (100%; due to a deprotonated galloyl residue loss with simultaneous decarboxylation and deprotonation) and secondary fragments at m/z 629 (25% of base peak; tetragalloylquinic acid − gallic acid − H_2_O − H^+^) and at m/z 477 (15%; tetragalloylquinic acid − (2 × gallic acid) − H_2_O − H^+^)). In the positive ion mode, the molecular ion [M+H]^+^ was detected at m/z 801, the loss of one gallic acid was observed in the MS^2^ spectrum where an ion fragment was present at m/z 631. A MS^3^ fragmentation of this base peak produced a fragment ion at m/z 461, which produced a base peak at m/z 291 in MS^4^. All of these fragmentations were due to three gallic acid losses and dehydrations.

Compounds **1–4** are digalloyl, trigalloyl, and tetragalloyl quinic acids. This is the first time that they have been described in *Tapirira guianensis.* Other hydrolysable tannins have already been found in previous studies: 1,4,6-tri-*O*-galloyl-D-glucose and galloyl-HHDP-hexoside [13].

Compound **5** showed a UV spectrum with a maximum at 279 nm and a molecular ion at m/z 441 in the negative ion mode. The MS^2^ base peak ion at m/z 289 was due to a loss of 152 (loss of one galloyl residue). Minor peaks were also observed at m/z 169 (gallate ion) and m/z 125 (decarboxylated gallate ion), consistent with an (epi)catechin gallate structure.

In the positive ion mode, a molecular ion was detected at m/z 443, and MS^2^ fragmentation yielded a base peak at m/z 273, corresponding to the loss of a galloyl residue and a water molecule (170 amu). The structure was definitively confirmed by comparison with standard epicatechin 3-O-gallate.

Compounds **6** to **16** displayed UV-visible spectra that are typical of flavones or flavonols, with two maxima between 265–280 nm (band II) and 346–371 nm (band I).

The UV spectrum of compound **6** had two wavelength maxima at 266 nm and 352 nm.

In the positive mode, the compound was detected at m/z 465 and demonstrated the same neutral loss (146 amu), which could be attributed to a deoxyhexoside residue that was observed with a peak at m/z 319 in MS^2^. MS^3^ fragmentation yielded peaks at m/z 301 (59% of base peak; M + H − H_2_O), m/z 291 (10%; M + H − CO), m/z 273 (base peak; M + H − H_2_O − CO), m/z 263 (24%; M + H − 2 CO), and m/z 245 (33%; M + H − H_2_O – 2 CO) [25]. As such, compound 6 was tentatively identified as being a myricetin deoxyhexoside. Other authors have described the presence of myricetin deoxyhexoside [13] in *T. guianensis* leaves and have identified it as myricetin 3-*O*-*α*-L-rhamnopyranoside (myricitrin) [14].

In ESI^-^, the parent ion at m/z 463 produced a MS^2^ base peak at m/z 317, corresponding to a 146 amu loss, which could be a deoxyhexoside residue. The ions resulting from MS^2^ fragmentation showed peaks at m/z 317 (base peak), m/z 316 (95% of base peak), m/z 271 (3%), and m/z 179 (6%). The ion fragmentation that occurred at m/z 316 was the result of the homolitic cleavage of the O-glycosidic bond, and the fragmentation at m/z 179 corresponds to retrocyclization after the loss of the B ring [26,27].

The UV spectra of products **7**, **9,** and **11** were almost identical. Parent ions were detected at m/z 433/435 (compounds **7** and **9**) and 447/449 (compound **11**) in negative/positive ion mode, respectively. All three yielded a fragment ion at m/z 301 in the negative ion mode and at m/z 303 in the positive ion mode, which could be attributed to quercetin resulting from the loss of a pentoside residue (132 amu) for compounds **7** and **9** and the loss of a deoxyhexoside residue for compound **11**. In the positive mode, MS^3^ fragmentation yielded peaks for the three compounds at m/z 285 (30% of base peak; M + H − H_2_O), m/z 275 (9%; M + H − CO), m/z 257 (base peak; M + H − H_2_O − CO), m/z 247 (60%; M + H − 2 CO), and m/z 229 (67%; M + H − H_2_O – 2 CO) according to Ma et al., 1997. Compounds **7** and **9** were attributed to quercetin pentosides, and compound **11** was attributed to a quercetin deoxyhexoside. A quercetin pentoside (quercetin- *α*-L-arabinofuranoside or avicularin) and quercetin deoxyhexose (quercetin-3-*O*-*α*-L-rhamnopyranoside or quercitrin) were previously described in the leaves of *TG* [14]. Compound **8** showed a UV spectrum with two wavelength maxima at 265 nm and 346 nm, which is typical of flavones and flavonols. In the positive ion mode, a parent ion was obtained at m/z 367 and fragmented in MS^2^ in a base peak at m/z 287 (corresponding to an 80 amu loss). This ion was then fragmented in MS^3^, leading to a typical fragmentation of kaempferol according to Ma et al., 1997 [25]. In the negative ion mode, the parent ion was detected at m/z 365 and fragmented in MS2 in an ion at m/z 285, corresponding to a loss of 80 amu. The hypothesis for why this occurred is that compound **8** may be a kaempferol derivative.

Compounds **10** and **12** displayed similar UV spectra with characteristic flavonol absorption. In ESI^+^, molecular ions appeared at m/z 449 and 463, respectively. MS^2^ base peaks were detected at m/z 317 and were determined to have been formed by the loss of pentoside (loss of 132 amu) and deoxyhexoside (loss of 146 amu). The MS^3^ fragmentation leading to peaks at m/z 302 (72% of base peak; M + H − CH_3_), 299 (6% of base peak; M + H − H_2_O), m/z 289 (2%; M + H − CO), m/z 285 (base peak; M + H − CH_3_OH), and m/z 257 (4%; M + H − CH_3_OH − CO) was consistent with that of isorhamnetin described by Ma et al., 1997. The ESI^-^ mass spectra were in accordance with losses of pentoside and deoxyhexoside, respectively, for compounds 10 and 12. Thus, compounds **10** and **12** were identified as isorhamnetin pentoside and isorhamnetin deoxyhexoside, respectively. To the best of our knowledge, this is the first report of isorhamnetin derivatives in the leaves of *Tapirira guianensis.*

Compounds **13** and **14** were attributed to kaempferol that had been glycosylated by either a pentosyl residue (loss of 132) or a deoxyhexosyl residue (loss of 146), respectively, regarding their mass fragmentations and according to Ma et al., 1997. Kaempferol-3-O-arabinoside (juglanin) and kaempferol-3-O-rhamnoside (afzelin) have already been described in literature by Correia et al., 2008 [15].

The T_R_, UV spectra, and mass fragmentations of **15** and **16** were identical to those of the standards for quercetin and kaempferol, respectively.

### 2.3. AO Activities in Crude Tapirira guianensis Extracts

Compared to the pure water or methanol extracts, the 85% MeOH extract showed better antiradical activity regardless of the antioxidant assay used (DPPH, TEAC, and ORAC) The results are shown in Table 3.

The antiradical activity of the *TG* hydromethanolic extract was similar to that of green tea (very deemed for its antioxidant properties). When we compared the *T. guianensis* H_2_O extract with the teas, we observed that *TG* (IC_50_: 15.9 ± 2.9 mg DE/L, data not shown) has lower DPPH activity than green tea (IC_50_: 4.14 ± 1.00 mg DE/L) but higher DPPH activity than black and oolong tea (IC_50_: 27.02 ± 0.96 and 47.12 ± 0.99 mg DE/L, respectively) [28]. In the ORAC assays, the *TG* methanolic and hydromethanolic extracts presented better antiradical activity than the green tea leaf extracts (577.49 ± 46.36 and 45.68 ± 6.22 μmol TE/g DM for 100% methanol and 50% methanol, respectively) [29].

The in vitro antioxidant chemical tests showed remarkable values, especially for the hydromethanolic extracts. These results demonstrated the necessity to assess a cell-based antioxidant assay to confirm the trends observed.

### 2.4. Fractionation of the 85% MeOH Extract

Following the results from above, partition was performed using 85% MeOH extract in order to determine which classes of polyphenolics were responsible for the antioxidant activity. The results are given in Table 4.

The total mass yield was quite high (88%), showing a good recovery after crude extract partition.

Fractions W1 and M1 were mainly composed of gallic acid, and both accounted for 44% of the crude extract yield. Highly hydrophilic compounds (such as mineral salts, free sugars, low molecular weight organic acids) are generally found when eluting with water. Galloyl quinic acids were only found in fraction W2 (9% of the crude extract weight), which also contained flavonols.

Flavonols and flavanols were mainly observed in fractions M2, W3, and M3 (35% of crude extract weight). Both contents accounted for more than 44% of the polyphenolics in the crude extract.

In terms of antiradical activities, the best results were obtained for fractions W2, W3, and M3 (56.5% in DPPH) in both assays, demonstrating the important role of flavonols. These metabolites (specifically quercetin, myricetin, and myricitrin) were already indicated to be bioactive compounds in previous studies [13,14].

Additionally, the AO activity in the W2 fraction was quite good in the DPPH assays. The activity in fractions W1 and M1 was low in the DPPH assay.

The antioxidant activity in *T. guianensis* antioxidant determined in the DPPH assay may be imputed to the total polyphenol contents since there is an excellent correlation coefficient (R^2^ = 0.97).

## 3. Materials and Methods

### 3.1. Chemicals

Milli-Q quality water, methanol, and formic acid were analytical grade-quality products purchased from Carlo Erba Reagents (Reuil, France). Folin–Ciocalteu reagent came from Carlo Erba reagents (Reuil, France). Fluorescein disodium salt, dimethylaminocinnamaldehyde (DMACA), and ABTS were obtained from Fluka- Sigma–Aldrich (Steinheim, Germany). The compounds 2,2-Diphenyl-1-picrylhydrazyl (DPPH), 6-hydroxy-2,5,7,8-tetramethylchromane-2-carboxylic acid (Trolox), 2,2′-azobis(2-methylpropionamidine) dihydrochloride (AAPH), and quercetin came from Acros Organics (Geel, Belgium). Gallic acid was purchased from Alfa Aesar (Ward Hill, MA, USA). Catechin came from Fluka Sigma-Aldrich (Steinheim, Germany). SPE cartridges (Strata) were purchased from Phenomenex (Torrance, CA, USA).

### 3.2. Plant Material and Extraction Procedures

Mature *T. guianensis* leaves were collected in Remire-Montjoly (French Guiana, France) in a sandbank forest during the dry season (September). The plant material was sequentially washed and dipped into liquid nitrogen to avoid enzymatic degradations of the polyphenols and ice crystal formation. Then, the samples were ground, freeze-dried, and stored until further analysis. The extraction method was adapted from Ummat et al., 2020, with some modifications [30]. Three polar solvents were selected to obtain the leaf extracts, and a green method was used (ultrasound-assisted extraction): pure methanol, pure water, and methanol/water (85/15, *v*/*v*). *TG* leaves (2.5 g) were extracted three times in 50 mL of each solvent under sonication (130 kHz, 10 min) at room temperature. Then, each extract was centrifuged (5000× *g*, 10 min), and the three supernatants were combined after filtration. Organic solvents were evaporated in a rotary evaporator under reduced pressure using a bath at 40 °C. Aqueous extracts were freeze-dried. Finally, the dried extracts were weighed to determine the dry extract yield and were redissolved in the original extraction solvent at a concentration of 40 mg/L.

### 3.3. Total Polyphenol, Flavonoid, and Flavanols Contents

All assays were performed in triplicate, and the standard deviation was calculated.

**The total polyphenol contents** (TPC) of the *T. guianensis* extracts were determined according to Arnous et al., 2002, [31] with some modifications [32]. Extracts with different concentrations (or reference or blank) (30 µL) were placed in glass tubes with water (2370 µL) and Folin–Ciocalteu reagent (150 µL), and they were then vortexed. An amount of 450 µL of Na_2_CO_3_ (20%; *w*/*v*) was added, and the solution was kept in the dark at room temperature for 2 h. Absorbance was read at 750 nm with a Cary 50^®^ Varian spectrophotometer (Agilent Technologies, Les Ulis, France). The gallic acid curve was calibrated using different concentrations (between 100 and 1000 mg/L) s as reference, and methanol was used as a blank. Results were expressed in mg gallic acid equivalents per g of dry matter (mg GAE/g DM).

**The total flavonoid contents** (TFC) were measured following the procedure from Kim et al., 2003, [33], with some modifications. Diluted extracts (or reference or blank) (400 µL) were placed in glass tubes with an aqueous NaNO_2_ solution (120 µL, 5% *w*/*v*). After 5 min, 120 µL of an aqueous AlCl_3_ solution (10% *w*/*w*) was added and vortexed. An amount of 800 µL of NaOH (1N) was added 1 min later, and then 960 µL of Milli-Q quality water was added quickly. The mixture was vortexed before the absorbance was read at 510 nm with a Cary 50^®^ Varian spectrophotometer (Agilent Technologies, Les Ulis, France). Different concentrations of catechin (between 20 and 120 mg/L) were used to calibrate the calibration curve. Results were expressed as mg catechin equivalent per g of dried matter (mg CE/g DM).

**The total flavanol contents** were determined according to Arnous et al., 2002, [31]. A dimethylaminocinnamaldehyde (DMACA) solution (0.1% *w*/*v*) was prepared in 1N HCl and kept in the dark. To 400 µL of the diluted extracts (or blank or standard), 2 mL of DMACA solution was added, vortexed, and then kept in the dark for 10 min. Absorbance was measured at 640 nm using a Cary 50^®^ Varian spectrophotometer (Agilent Technologies, Les Ulis, France). Contents were calculated using a calibration curve with different concentrations of catechin (between 3 and 15 mg/L). Results were expressed as mg catechin equivalent per g of dry matter (mg CE/g DM).

### 3.4. Fractionation of Crude Extract

Fractionation was performed by solid phase extraction (SPE) according to the procedures used by De Villiers et al., 2004, and Monagas et al., 2003, [34,35], with some modifications. The crude extract (10 mL) was dried and then partially dissolved in water (3 mL) and filtered (0.2 µm). The filtrate was the W fraction. The microfilter was washed with MeOH, and then the eluent was dried and dissolved in a MeOH (1.5 mL)/water (8.5 mL) mixture to acquire the M fraction. Each W and M fraction was put into a different Strata SPE cartridge containing 2 g of C18 silica that had previously be conditioned with MeOH (10 mL) and water (2 × 10 mL) and then submitted to successive partitions with three solvents in the following order: water (fraction 1), AcOEt (fraction 2), and MeOH (fraction 3). Thus, six fractions were sequentially obtained: W1, W2, W3 and M1, M2, M3 from the W and M fractions, respectively. All of the fractions were dried, re-suspended in 10 mL of either water (W1, W2 and W3) or methanol (M1, M2, and M3), and stored at −20 °C for UPLC analyses, the conditions of which are described in 3.5.

### 3.5. UPLC-DAD-ESI-IT-MS^n^ Analysis

Separation was performed according to Habib et al., with some modifications, on an UPLC Acquity^®^ system (Waters, Saint-Quentin-en-Yvelines, France) equipped with and a photodiode array detector (PDA) coupled to a BRUKER AMAZON X^®^ ion-trap mass spectrometer (Bruker France SAS, Champs sur Marne, France) using Electrospray Ionization (ESI). An Acquity UPLC^®^ BEH C18 reversed-phase column (1 mm × 150 mm, 1.7 µm, 130Å) from Waters was used at 35 °C [36]. The flow rate was set at 0.08 mL/min. The injected volume was 0.5 µL. Acidified water (1% formic acid, *v*/*v*) and acidified MeOH (1% formic acid, *v*/*v*) were used as mobile phases A and B, respectively. The elution gradient was 0 min 2% B; 1 min 2% B; 10 min 30% B; 12 min 30% B; 25 min 75% B; 30 min 90% B; 35 min 90% B; and 38 min 2% B. Detection was registered in the 280, 320, 360, and 520 nm UV/Vis regions. The mass spectrometer parameters were set as follows: HV capillary voltage ± 2.5 kV, nitrogen as nebulizer and drying gas; nebulizer pressure of 14.5 psi; drying gas with flow rate of 10 L/min; and a set temperature to 200 °C. Mass spectra were obtained over a range of m/z 400–1500 operating in positive and negative ion modes.

### 3.6. DPPH, TEAC and ORAC Assays

Chemical antioxidant properties were assessed using the DPPH, TEAC, and ORAC assays. These assays involve different antioxidant action modes such as hydrogen atom transfer (ORAC) or single electron transfer (DPPH and TEAC) [37]. Chemical substances (Trolox) and plant extracts (green tea) were used as references.

**The DPPH assay** was implemented according to Kordali et al., 2005, with some modifications [38]. An amount of 100 µL *T. guianensis* extract that had been diluted at different concentrations was mixed with 3900 µL methanolic 0,1 mM DPPH solution and kept in a dark place for 90 min. Absorbances were measured at 515 nm with a Cary 50^®^ Varian spectrophotometer (Agilent Technologies, Les Ulis, France) using methanol as a blank. DPPH scavenging activity was expressed in µmol Trolox equivalent per g of dry matter (µmol TE/g DM). The percentage of DPPH free radical inhibition was evaluated by following the below equation:% Inhibition = (ODc − ODs) × 100%/ODc
where ODc is the OD (absorbance) value of the negative control (blank), and ODs is the OD value of the testing sample. The IC50 value indicating the concentration at which a sample would inhibit free radicals by 50% was also calculated and expressed in mg/L.

**The TEAC assay** is based on the reduction of the 2,2′-azinobis (3-ethylbenzothiazoline-6-sulphonate) radical cation (ABTS+.) to colorless ABTS. For this test, the method of Re et al., 1999, was followed with some modifications [39]. The radical cation was pregenerated with the addition of 10 mL potassium persulfate solution (4.9 mM) to 10 mL ABTS methanolic solution (14 mM) and was kept in the dark for 16 h. An amount of 30 µL methanolic T. guianensis extract solution at different concentrations was added to 2970 µL of activated pregenerated ABTS solution and was kept in a dark place at room temperature for 90 min. Then, absorbance reduction was recorded at 734 nm on a Cary 50^®^ Varian spectrophotometer (Agilent Technologies, Les Ulis, France) using distilled water as a blank. Results were given in µmol Trolox equivalent/g of dry matter (µmol TE/g DM).

**The ORAC assay** was conducted according to the method of Ou et al., 2001, with some modifications [40]. Analyses were performed in phosphate buffer pH 7 (75 mM) at 37 °C. Peroxyl radical was generated using 2,2′-azobis (2-amidino- propane) dihydrochloride (AAPH), which was freshly prepared for each run. Fluorescein (FL) was used as a substrate. First, the samples (at different concentrations) or blank (methanol) were mixed with phosphate-buffered solution and fluorescein. The mixture was then preincubated for 10 min at 37° in an Eclipse Varian^®^ fluorescence spectrophotometer (Agilent Technologies, Les Ulis, France). AAPH solution was added, and fluorescence was recorded at 1 min intervals for 70 min until it reached less than 5% of the initial intensity (excitation wavelength 485 nm, emission wavelength 520 nm). Measurements were conducted in quadruplicate. Results were calculated based on the area differences of the fluorescein decay curve between the blank and the sample. They were expressed as µmol Trolox equivalent/g of dry matter (µmol TE/g DM).

### 3.7. Statistical Analysis

Statistical analysis was conducted STATA\IC-version 12 software and was determined using non-parametric analysis (Mann–Whitney U test). Significance was accepted at the 5% level (*p* ≤ 0.05).

## 4. Conclusions

*Tapirira guianensis* hydromethanolic (85%) leaf extract produced the highest yield and total polyphenol content. This was not the case for the total flavonoid and total flavanol contents, as generally, other extraction solvents (acetone for instance) are more required. The FRS content of TG was described for the first time using spectrophotometry. However, an accurate analysis using chromatography is needed to confirm these first results.

Two main classes of polyphenols (galloylquinic acids and flavonols) were evidenced in the hydromethanolic extract, regardless of whether the structural elucidation of the polyphenolics must be confirmed using ^13^C and ^1^H NMR in order to clarify the nature of the sugar and its position of the flavonoids aglycone and galloyl quinic acids. Given that our study was focused on hydromethanolic extract compared to other extracts, galloylquinic acid derivatives were described in TG for the first time. They have already been reported as metabolites with biological properties that include antioxidant, leishmanicidal, anti-HSV-1, and anti-allergy bioactivities [41,42]. Our work also highlights the presence of isorhamnetin for the first time. Further quantifications of these compounds should be performed.

The best antioxidant activity was determined in TG hydromethanolic extract when the three chemical AO assays were used. The DPPH, TEAC, and ORAC data were similar to (or better than) to those of green tea, which is often recommended due to its properties. Flavonols and galloyl quinic acids seem to play a major role in DPPH antiradical activity. They have also been mentioned by previous authors as being key bioactive molecules with anti-tumoral, vasodilatory, and myeloperoxidase-inhibitory properties. Further studies dealing with antioxidant activity on cells and other biological assays could be of great interest.

In conclusion, these findings provide a rationale for the use of *Tapirira guianensis* in addition to traditional medicine. This Amazonian tree could be a new source of valuable bioactive compounds for cosmetics or nutraceuticals.

## Figures and Tables

**Figure 1 plants-11-00326-f001:**
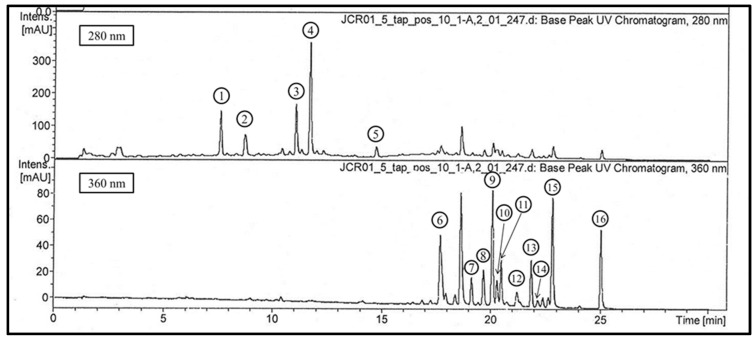
LC-UV chromatogram of the crude extract from *T. guianensis* leaves (methanol/water; 85/15, *v*/*v*).

**Table 1 plants-11-00326-t001:** Yield and composition of *T. guianensis* extracts.

Sample	Extraction Yield ^1^ (%)	Total Polyphenol Contents ^2^ (mg GAE/g DM)	Flavonoid Contents ^3^ (mg CE/g DM)	Flavanol Contents ^3^ (mg CE/g DM)
H_2_O extract	28.0 ± 1.6 ^a^	40.5 ± 14.2 ^a^	16.7 ± 6.3 ^a^	14.7 ± 8.6 ^a^
MeOH extract	28.9 ± 0.4 ^a^	69.2 ± 1.9 ^b^	25.6 ± 0.8 ^b^	43.0 ± 5.5 ^b^
MeOH/H_2_O (85/15) extract	28.1 ± 1.6 ^a^	112.7 ± 5.8 ^c^	17.9 ± 0.5 ^a^	49.0 ± 5.6 ^b^

^1^ Percentage related to dry matter. ^2^ Expressed in mg gallic acid equivalents per g of dry matter (mg GAE/g DM). ^3^ Expressed in mg catechin equivalent per g of dry matter (mg CE/g DM). All results are given with standard deviation (±SD). Different letters in the same column indicate a significant difference according to the Mann–Whitney U-test.

**Table 2 plants-11-00326-t002:** UPLC-DAD-IT-MS^n^ fragmentation data (in positive and negative modes) for the 85% methanolic extract of *T. guianensis* (relative ion intensity in brackets).

Product	t_R_	λ_max_ (nm)	[M−H]^−^ (m/z)	MS^2^ (m/z) Negative Mode	[M+H]^+^ (m/z)	MS^2^ (m/z) Positive Mode	MS^3^ (m/z) Positive Mode	Structural Hypothesis (Class)	Sources
1	7.7	280	495	343(100) 325(2) 191(3) 169(6)	497	479(100)	309	3,5-Digalloylquinic acid	24
2	8.8	280	495	343(100) 325(38) 193(18) 169(14)	497	479(100)	309	4,5-Digalloylquinic acid	24
3	11	275	647	495(100) 477(13) 343(12) 325(7)	649	479(100)	309	3,4,5-Trigalloylquinic acid	24
4	11.6	275	799	601(100) 629(32) 477(12)	801	631 461	461 (from m/z 631) 291(from m/z 461)	1,3,4,5-Tetragalloylquinic acid	24
5	14.7	279 (tailling)	441	289(100) 169(11) 125(4)	443	273		epicatechin gallate (flavanol)	
6	17.7	266 352	463	317(100) 316(95) 271(3) 179(6)	465	319	301(59) 291(10) 273(100) 263(24) 245(33)	Myricetin deoxyhexoside (flavonol)	14, 26, 27
7	19.1	256 353	433	301(100)	435	303	285(30) 275(9) 257(100) 247(60) 229(67)	Quercetin pentoside (flavonol)	14
8	19.6	265 346	365	285	367	287	269(77) 259(56) 231(22) 213(61)	Kaempferol derivative * (flavonol)	25
9	20.1	256 353	433	301(100)	435	303	285(30) 275(9) 257(100) 247(60) 229(67)	Quercetin pentoside (flavonol)	14
10	20.4	265 355	447	315	449	317	302(72) 299(6) 289(2) 285(100) 257(4)	Isorhamnetin pentoside (flavonol)	25
11	20.6	265 350	447	301	449	303	285(30) 275(9) 257(100) 247(60) 229(67)	Quercetin deoxyhexoside (flavonol)	14
12	21.3	265 350	461	315	463	317	302(72) 299(6) 289(2) 285(100) 257(4)	Isorhamnetin deoxyhexoside (flavonol)	25
13	1.8	264 371	417	285	419	287	269(77) 259(56) 231(22) 213(61)	Kaempferol pentoside (flavonol)	15
14	22.2	264 368	431	285	433	287	269(77) 259(56) 231(22) 213(61)	Kaempferol deoxyhexoside (flavonol)	15
15	22.8	255 371	301		303	285(30) 275(9) 257(100) 247(60) 229(67)		Quercetin (flavonol)	
16	25	265 365	285		287	269(77) 259(56) 231(22) 213(61)		Kaempferol (flavonol)	

* coelution; t_R_ retention time (min).

**Table 3 plants-11-00326-t003:** Antiradical activity of *Tapirira guianensis* extracts.

Sample	DPPH Assay ^1^ (µM TE/g DM)	TEAC Assay ^2^ (µM TE/g DM)	ORAC Assay ^3^ (µmol TE/g DM)
H_2_O extract	437.5 ± 77.1 ^a^	356.4 ± 77.3 ^a^	597.2 ± 114.5 ^a^
MeOH extract	817.2 ± 101.1 ^b^	708.8 ± 102.5 ^b^	779.2 ± 148.9 ^a^
MeOH/H_2_O extract	1050.4 ± 21.9 ^c^	938.7 ± 110.4 ^b^	2567.3 ± 476.3 ^b^
Green tea extract ^3^	969.7 ± 54.3	822.6 ± 87.6	2911 ± 221.7

^1^ Expressed in mg of dried extract (DE)/L. ^2^ Expressed in µmol Trolox equivalent (TE)/g of dry matter (DM). ^3^ Hydromethanol (85%) was used as solvent. All results are given with standard deviation (±SD). Different letters in a same column indicate a significant difference according to the Mann–Whitney U-test.

**Table 4 plants-11-00326-t004:** Hydromethanolic extract fractionation. Yield, total polyphenol content, and antiradical activities in DPPH assays.

Fractions	Mass Yield (%) ^1^	Total Polyphenol Contents ^2^	DPPH Assay ^3^
Crude extract	100%	335	100%
W1	27%	9.5	3%
M1	17%	16.5	5.5%
W2	9%	64	20%
M2	14%	14.5	3.5%
W3	16%	96	27%
M3	5%	37	9.5%
Total	88%	237	69%

^1^ Compared to initial extract. ^2^ Expressed in mg gallic acid equivalents per g of dry matter (mg GAE/g DM). ^3^ Role in DPPH inhibition (%).

## Data Availability

The data presented in this study are available on request from the corresponding author. The data are not publicly available due to it is not in an online archive.

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
