# Peer review of "Polyphenol Composition and Antioxidant Activity of Tapirira guianensis Aubl. (Anarcadiaceae) Leaves"

_plants, 2022, doi:10.3390/plants11030326_

Round 1

Reviewer 1 Report

The paper titled with “Polyphenol composition and antioxidant activity of Tapirira 2 guianensis Aubl. (Anarcadiaceae) leaves" by Arnaud Patient et al, describes AO assays (DPPH, TEAC, ORAC) implying T. guianensis leaf extracts obtained with water, methanol, and 85%-methanol. Polyphenol and flavonoid contents were determined spectrophotometrically (UV/Visible). Composition of polyphenol extracts was also determined by UPLC-DAD-ESI-IT-MSn.

General comments as follows

1)            Pag 4 Figure1: why did you not characterize the signal between 6 and 7 peaks despite being very intense?

2)            Pag. 4 Table 2: to be clearer a column with bibliographical references should be added to the table.

3)            Pag 9 line 273: reference 27 is not correct, it is not relevant.

4)            Pag 9 Table 3 DPPH essay should be expressed in % of inhibition.

5)            Pag 10 Line 302 the sum of W2, W3 and M3 values is not 40 % in DPPH. 

6)            Pag 11 line 375-376: Total Phenol contents (TPC): indicate if you have constructed a calibration standard curve

7)            Pag 12 paragraph 4.4:  add a sentence about UPLC analysis.

8)            Ref 4; Ref 5: please replace or add with others one in English. Rif 6 add book references

Once these questions are addressed, the article should be published.

Reviewer 2 Report

The novelty of this research relies on the study of an exotic and abundant plant species (Central and South America)  that can be valorized due to the presence of bioactive compounds in their leaves.

Green technologies and solvents were used.

Some inconsistent designations were found.

Other comments and suggestions are in the attached word file

Author Response

This manuscript is a resubmission of an earlier submission. The following is a list of the peer review reports and author responses from that submission.

Round 1

Reviewer 1 Report

In this paper by Patient et al., the Authors investigated the phenolic composition of T. guianensis leaves using three different extraction solvents. Moreover, the different extractions were used to assess their cyto-protective effects in fibroblast in culture.

The paper is very bad written and contains many grammatical and editing mistakes. This makes the paper very difficult to read. Beside it, the experimental design is not appropriate, the methods are not well explained making impossible to reproduce them and the experiments on cell culture are not well conducted.

Moreover:

The title is too generic and does not fit with the aim of the study

Statistical analysis is missing

Tables. Standard deviation is missing.

Tables. What does it mean the number among brackets? If it is the standard deviation it should be better to use ±. Anyway explain it in table legends

The FHN/HXXO assay is not present in literature and it is not clear what means “described by Silab”. Was the kit provided by Silab? Was the assay performed by an Author affiliated at Silab? It is very different and not clear. This method is not appropriately described. No references are present

Cell culture maintaining protocol should be included.

It is not clear what the Authors explain in paragraph 4.8. MTT assay measure the conversion of MTT to water insoluble salt of formazan. After incubation with MTT, formazan is dissolved in a proper solvent and quantified by a spectrophotometer. The method described does not correspond to the MTT assay but seems more related to trypan blue exclusion test.

Data availability statement: delete template statements

Reviewer 2 Report

The manuscript entitled “Polyphenolics study of Tapirira guianensis Aubl. (Anarcadiaceae) leaves and antioxidant activity assessment using chemical and biological tests” (Manuscript ID: plants-1098037) deals about antioxidant properties of extract obtained from Tapirira guianensis. The article is good organized and presented. However, it has weaknesses and is lacking of information that have to be greatly improved. The name of research material should be italic write in all manuscript.

Abstract

Line 25. The AO abbreviation should be clarified

Introduction

Lines 77 and 96: What does it mean “…..”?

Results

Table 1. The statistic method is missing. Was the standard deviation as single-number?

Line 99. Initials should be deleted.

Table 3. If you tested extracts the antioxidant activity should be show as a EC50 value.

The statistical method is missing.

Table 4. Was the sample concentration the same in all tests?

Line 231. There are only conjectures. In my opinion into prepare conclusion you should compare the results with pure compounds that was indentified in extracts.

Discussion section is weak and should be rewritten.

Reviewer 3 Report

The manuscript “Polyphenolics study of Tapirira guianensis Aubl. (Anarcadiaceae) leaves and
antioxidant activity assessment using chemical and biological tests” provides insight knowledge
about antioxidant activities of Tapirira guianensis Aubl. However, the main objective of this
study is still unclear which led this experiment and its future prospects as well. The methodology
section is also confusing regarding any specific treatments given to derive this experiment.
Overall, this manuscript is well written and thoroughly described. However, there are a lot of
English language and sentence errors in overall writeup that cannot be overlooked at this stage.
Therefore, editing of English language and style is required by some English mother tongue
expert. Moreover, the whole manuscript is not properly formatted as per author’s guidelines
provided by the journal, particularly references section.
Some other major comments are as follows
The title is confusing and should be revised in a proper scientific manner.
There are a lot of abbreviations used throughout the manuscript, in this way it is suggested to
provide a separate list of abbreviations before the introduction section.
Mind the space between lines 29 and 30.
The keywords must be different from title words
The introduction section needs to be improved properly. It describes the importance of Tapirira
guianensis Aubl crop but does not speak anything about its area and production data worldwide.
The main scope and future prospects of this study must also be included in the introduction
section with some latest literature. Moreover, there are a lot of English language mistakes in this
section as well. The introduction section includes most of the irrelevant literature keeping in
view the main objectives of the study even the references in the text are not properly formatted.
Material and methods: I don’t think first heading of chemicals is necessary. The methodology
section is very confusing. The major contradiction arises here when there is no such treatment
and any other factor was involved in this study which makes it unclear about statistical
significance of the results, that draws down the significance of this study.
Reference missing for the procedures adopted in section 4.2, line 267. Same is the case for 4.5.
UPLC-DAD-ESI-IT-MS n analysis
The second major drawback of the manuscript is the results and discussions section which lacks
the solid justifications and reasons for the results obtained. There are some irrelevant and
outdated literature references that does not suit to the arguments provided by the author. Most of
the results are too confusing and exaggerated which limits the importance of the manuscript.
Moreover, the English language also needs some serious improvements.
Line 244: 3. Discussions or conclusions???
The references are not properly formatted, nor in the text, neither in the citations. Please refer to
Author’s guidelines and revise the references accordingly.

In summary, presented data show a positive scenario regarding antioxidants activities, but
discussion of results is very speculative. To sum up, the manuscript can find interest among
specialists when these major comments will be taken into account. However, in the present form,
I cannot recommend the work for acceptance.

Reviewer 4 Report

The manuscript enlisted “Polyphenolics study of Tapirira guianensis Aubl. (Anarcadiaceae) leaves and antioxidant activity assessment using chemical and biological tests”, authored by Patient and colleagues, deals with the profiling of phytochemicals present in the extracts from Tapirira guianensis. Moreover, the authors investigated the bioactivity of the extracts, measuring both the antioxidant properties (DPPH, TAE and ORAC), biological activity (FHN/HXXO) and cytotoxicity. The manuscript contains really interesting data that can find a useful application for new sustainable approaches. However, before consider suitable the manuscript for the publication, some changes need to be addressed, especially in Discussion section.

LINE 10: there is a letter coloured in red.

Please, check that Tapirira guianensis, as well as T. guianensis, is written in italic all over the text.

Please, check that Anarcadiaceae is written in italic all over the text.

In Affiliation section, the author’s acronym should be reported after each e-mail address. Moreover, check that the e-mail addresses of each authors are correctly reported in the affiliation section.

Keywords should be words not contained in the title of the manuscript. Their usefulness is making the manuscript more available and researchable using the universal search engines (PubMed). Since many keywords are already contained in the title, I suggest to replace them and to add new ones, but respecting the limit provided by the journal.

The use of plant-based extracts is an important topic in the last years. In particular, plants are actually used for the extraction of important bioactive compounds that may be useful to guarantee the human well-being and for the preparation of supplements or nutraceuticals enriched in these compounds (https://doi.org/10.3390/nu12040992). Furthermore, thank the valorisation of plants as sources of bioactive compounds, the market of dietary supplements, and in general of nutraceuticals, is quickly growing. In this context, the use of plant leaves as alternative sources for the extraction of phytochemicals is a sustainable practice which is also growing a lot in recent years. In particular, the authors could also mention that the leaves are actually considered an agricultural waste, which can be revaluated for the extraction of bioactive compounds or for the formulation of supplements rich in polyphenols that may be useful for the prevention or treatment of several diseases (https://doi.org/10.3390/antiox9020101; https://doi.org/10.3390/molecules25112612).

In table 1, 2, 3 and 4 the values should be reported using the same significative figures (28.2 ± 0.3).

In table 3 and 4 the standard deviations of the most part of the calculated values are missing.

The section relating to the discussion is very little developed. In particular, the authors could compare the data obtained during their experiments with those reported in the literature. For example, many of the compounds identified and quantified by the authors are ubiquitously present in the leaves of many other genera (https://doi.org/10.3390/molecules25112612; https://doi.org/10.1371/journal.pone.0232599; https://doi.org/10.3390/plants9040442; https://doi.org/10.3390/agronomy11010089).